# BcXyl, a β-xylosidase Isolated from *Brunfelsia Calycina* Flowers with Anthocyanin-β-glycosidase Activity

**DOI:** 10.3390/ijms20061423

**Published:** 2019-03-21

**Authors:** Boyu Dong, Honghui Luo, Bin Liu, Wenjun Li, Shaojian Ou, Yongyi Wu, Xuelian Zhang, Xuequn Pang, Zhaoqi Zhang

**Affiliations:** 1State Key Laboratory for Conservation and Utilization of Subtropical Agro-bioresources/Guangdong Provincial Key Laboratory of Postharvest Science of Fruits and Vegetables/College of Horticulture, South China Agricultural University, Guangzhou 510642, China; kuroro1986@163.com (B.D.); hhluo@stu.scau.edu.cn (H.L.); bin278083@foxmail.com (B.L.); lookitso@hotmail.com (W.L.); xuelianzhang@scau.edu.cn (X.Z.); 2College of Life Sciences, South China Agricultural University, Guangzhou 510642, China; osj1991@163.com (S.O.); zoewu09@foxmail.com (Y.W.)

**Keywords:** *Brunfelsia calycina*, anthocyanin degradation, β-xylosidase, anthocyanin-β-glycosidase, gene expression

## Abstract

*Brunfelsia calycina* flowers lose anthocyanins rapidly and are therefore well suited for the study of anthocyanin degradation mechanisms, which are unclear in planta. Here, we isolated an anthocyanin-β-glycosidase from *B. calycina* petals. The MS/MS (Mass Spectrometry) peptide sequencing showed that the enzyme (72 kDa) was a β-xylosidase (BcXyl). The enzyme showed high activity to p-Nitrophenyl-β-d-galactopyranoside (pNPGa) and p-Nitrophenyl-β-d-xylopyranoside (pNPX), while no activity to p-Nitrophenyl-β-d-glucopyranoside (pNPG) or p-Nitrophenyl-β-D-mannopyranoside (pNPM) was seen. The optimum temperature of BcXyl was 40 °C and the optimum pH was 5.0. The enzyme was strongly inhibited by 1 mM D-gluconate and Ag^+^. HPLC (High Performance Liquid Chromatography) analysis showed that BcXyl catalyzed the degradation of an anthocyanin component of *B. calycina*, and the release of xylose and galactose due to hydrolysis of glycosidic bonds by BcXyl was detected by GC (Gas Chromatography) /MS. A full-length mRNA sequence (2358 bp) of *BcXyl* (NCBI No. MK411219) was obtained and the deduced protein sequence shared conserved domains with two anthocyanin-β-glycosidases (Bgln and BadGluc, characterized in fungi). BcXyl, Bgln and BadGluc belong to AB subfamily of Glycoside hydrolase family 3. Similar to *BcPrx01*, an anthocyanin-degradation-related *Peroxidase* (*POD*), *BcXyl* was dramatically activated at the stage at which the rapid anthocyanin degradation occurred. Taken together, we suggest that BcXyl may be the first anthocyanin-β-glycosidase identified in higher plants.

## 1. Introduction

Anthocyanins are some of the most important plant natural pigments, responsible for a wide range of colors from red to purple and blue. The anti-oxidant activities of anthocyanins, eliminating excess free radicals, protect the plant tissues and may be of benefit for human health as natural food colorants [1]. The types and the amount of anthocyanins in vivo determine both the color quality and nutrient values of horticultural crops [2,3]. In contrast to the amount known on their biosynthesis, very little is known about the degradation process of anthocyanins in living plant tissues. 

Decreases of anthocyanin content are found in the plant tissues at specific developmental stages or due to changes in environmental conditions [4]. Rapid loss of anthocyanin occurs in many horticultural crops, for example, *Brunfelsia calycina* and *Rosa hybrida* petals after flower opening [5,6], fruit maturation of red apple and pear cultivars [7], and senescence of Litchi [8,9] and blood citrus [10] fruits, leading to quality reduction. Studies in fruit extracts and juices provided initial clues regarding the *in planta* anthocyanin degrading enzymes. Two enzyme families were shown to be involved in the in vitro process in fruit juices: polyphenol oxidases (PPO), and β-glycosidases [4]. One possible pathway for enzymatic anthocyanin degradation is PPO-phenols-anthocyanins system, in which PPOs first catalyzed the oxidation of phenols to form quinones, which were unstable molecules prone to form polymers with anthocyanins, leading to decoloration, browning and reduced anthocyanin content. Another possibility is two-step degradation, first deglycosylation of sugar moieties by β-glycosidases and then oxidation of the aglycone by PPOs [4]. Since PPOs are not present in the vacuoles but in the cytosol, it was suggested that the enzymes only functioned when the cells were de-compartmented. Some β-glycosidases were found to be located in the vacuoles and were suggested to be involved in *in planta* anthocyanin degradation [4].

Recently, we identified that a laccase (ADE/LAC, an anthocyanin degradation-related laccase) was responsible for the anthocyanin degradation during Litchi pericarp browning. ADE/LAC was demonstrated to be located in vacuole and degrade the pigment in an ADE/LAC epicatechin coupled oxidation model similar to the PPO-phenol-anthocyanin model. It shared some substrate specificity range with PPOs, such as catechol and 2,2’-Azinobis-(3-ethylbenzthiazoline-6-sulphonate) (ABTS) [11]. In addition, active enzymatic anthocyanin degradation dependent on novel mRNA and protein biosynthesis was found in *Brunfelsia calycina* flower petals, in which a basic Class Ⅲ peroxidase, BcPrx01, was found to be responsible for the *in planta* degradation of anthocyanins [5,12]. BcPrx01 localizes in the vacuoles of petals and has the ability to degrade complex anthocyanins [12]. Taken together, LACs and PODs are the two enzyme families, that were demonstrated to colocalize with the pigments in the vacuoles and be involved in the *in planta* anthocyanin degradation so far.

Like many secondary metabolites, anthocyanins exist as stable glycosylated compounds. The biological functions of these secondary metabolites are dependent on the liberation of the aglycones via the hydrolysis of the glycosidic bonds by β-glycosidase. The conjugate-hydrolyzing-glucosidase isolated from soybean (*Glycine max* (L.) Merr.) can directly hydrolyze genistein 7-*O*-(6″-*O*-malonyl-*β*-d-glucoside) to produce free genistein, which as chemical attractants play a very important role in the interaction of plants with microorganisms [13]. In addition, many previous studies on legumes have demonstrated that these flavonoids obtained by hydrolysis via glycosidase play an important role in the defense against pathogen infection [14,15,16]. In high anthocyanin content plant tissues, β-glycosidases may also be responsible for the hydrolysis of glycosidic bonds to liberate the anthocyanidins and sugars. Some species of fungi were found to contain high anthocyanin β-glycosidase activities [17]. A wine yeast expressing a glycosidase (Bgln) from *Aspergillus niger* fungi was found to have anthocyanin-β-glycosidase activities and was used to produce white wine from red grape muds [18,19]. An anthocyanin-β-glucosidase was found in blood orange, showing activities on the synthetic β-glycoside-containing substrate p-nitrophenyl-β-d-galactopyranoside (pNPGa) and on the anthocyanins in the fruit [10]. Up to now, however, no high-purity anthocyanin-β-glucosidases or the genes encoding the enzymes have been obtained from plants, and whether there are glycosidases that degrade anthocyanins in plant tissues or not is still unclear.

Previous studies have obtained evidence of enzymatic anthocyanin degradation in the flowers of *B. calycina* [12]. The color of the *B. calycina* petals changed from deep purple to pure white within 2 days of opening, at a specific and distinct stage of flower development, making this unique system well suited for studying the enzymatic process of anthocyanin degradation [5,12]. Concomitant with the pigment degradation, *B. calycina* flowers undergo other changes such as emission of benzenoid fragrances [20]. It is well known that the hydrolysis of glycosylated aroma by β-glycosidase is required for the release of the benzenoid fragrance, suggesting that high activity of β-glycosidase may function on both anthocyanin and aroma precursors in the flowers of *B. calycina*. In our previous study, it was found that Litchi PODs showed higher activities to the anthocyanidin aglycones than to their glycosides [9]. Whether β-glycosidases may also function in the hydrolysis of glycosylated anthocyanins to benefit the rapid pigment degradation in the flowers of *B. calycina* is unclear, and no information about β-glycosidases in the flowers of *B. calycina* has been reported.

Here, to investigate whether β-glycosidases may be involved in the rapid anthocyanin degradation in *B. calycina* flowers, we isolated β-glycosidases from *B. calycina* petals at the color fading stage by protein purification and sequencing. A β-xylosidase (BcXyl) was obtained and showed anthocyanin-β-glycosidase activity, and a gene coding for BcXyl was cloned and showed high expression level in the petals at the rapid color change stage.

## 2. Results

### 2.1. High Level of β-Glycosidase Activity in Brunfelsia Calycina Flower Petals during Development

*B. calycina* flower buds are dark purple with large amounts of anthocyanins. After the flower completely unfurls, the purple petals rapidly become white in three days (Figure 1A). To investigate the rapid pigment degradation, we divided the flower development into four stages, with flower bud as stage 1, completely-unfurled purple, light purple, and white color flowers as stage 2, 3 and 4 respectively (Figure 1A). Large pigment degradation was detected from stage 2 to 4, with anthocyanin content decline from 0.415 ± 0.027 to 0.051 ± 0.009 mg/g FW (Fresh Weight) (Figure 1C). 

To investigate whether β-glycosidase is involved in the anthocyanin degradation, we incubated p-Nitrophenyl β-d-galactopyranoside (pNPGa) with the crude enzyme extracts of the petals. After 30 min of incubation, the reaction mixture turned obvious yellow, indicating p-Nitrophenol was liberated and β-glycosidase activity existed in the petal tissue (Figure 1B). Slightly higher β-glycosidase activity was detected at stage 2 to 4, when compared to stage 1 (Figure 1D).

### 2.2. A β-Glycosidase was Purified from B. calycina Petals

Purification of β-glycosidase from *B. calycina* petals was achieved by ammonium sulfate fractionation, DEAE (Diethylaminoethyl)-Sepharose, and CM (Carboxymethyl)-Sepharose chromate-graphy (Table 1). About 50% of the total activity, detected with pNPGa as a substrate, remained in the fraction after ammonium sulfate precipitation. The enzyme was further purified by DEAE-Sepharose column chromatography and one major activity peak was obtained, partially overlapped with the main protein peak (Figure 2A). The eluted fractions with high activity in the activity peak (Tube No. 18–26) were pooled, and further subjected to CM-Sepharose column. One major activity peak was detected after the CM-Sepharose chromatography, overlapping with one protein peak in the fractions (Figure 2B), and the fractions with high activity in the major activity peak (Tube No. 26–28) were pooled. The specific activity of the combined fraction was 318.57 μmol/(h·mg protein) and a 44.56-fold higher purity was obtained after the sequential steps of purification (Table 1). The protein profile of the purified fraction after CM-Sepharose chromatography was visualized by SDS-PAGE (sodium dodecyl sulfate polyacrylamide gel electrophoresis), and a major band of approximately 72 kDa was observed (Figure 2C), indicating that the enzyme protein was of high purity.

### 2.3. The Purified β-Glycosidase was a β-Xylosidase

The major band present in the final purified fraction was sequenced by tandem mass spectrometry (MS/MS). Five peptides with high similarity to published protein sequences were identified (Table 2). Among these peptides, four peptides (peptide 1, 3, 4, 5, Table 2) were identical to *Solanum tuberosum* beta-xylosidase/alpha-L-arabinofuranosidase (β-Xyl/α-l-Afa) (XP_006351808.1). Peptide 1, 2, 3 showed identity with *Nicotiana tomentosiformis* β-Xyl/α-l-Afa (XP_009612011.1) and peptide 1, 2, 3, 5 were identical to *Nicotiana sylvestris* β-Xyl/α-l-Afa (XP_009762535.1) (Table 2). Based on the above results identified by the protein MS/MS spectra, and the following enzyme specificity analysis (Table 3), we predicted that the purified enzyme was a beta-xylosidase, and designated it as BcXyl.

### 2.4. Enzymatic Properties of the β-Xylosidase (BcXyl)

The pH optimum for the beta-xylosidase (BcXyl) activity was 5.0 and the activity decreased around 45% and 62% when the pH was lowered to 4.0 or increased to 6.0 respectively (Figure 3B). The temperature optimum for BcXyl activity was detected to be 40 °C. Around 40% of the activity at 40 °C was retained when the temperature increased to 70 °C (Figure 3A).

In order to further understand the enzymatic properties of BcXyl, its activity was measured with four different substrates under the same reaction conditions. The results showed that the enzyme showed the highest activity to pNPGa, with the activity as 301.35 ± 4.1 μmol/h/mg protein (Table 3). The enzyme also exhibited activity with pNPX (p-Nitrophenyl β-d-xylopyranoside), with 39.36% of the activity with pNPGa (Table 3). The enzyme showed no activity to pNPM (p-Nitrophenyl β-d-mannopyranoside) and pNPG (p-Nitrophenyl β-d-glucopyranoside). The substrate specificity of BcXyl indicates that the enzyme showed different activity to the substrates with different hexose moieties.

The effect of several substances on BcXyl was also investigated. As shown in Table 4, d-gluconic acid, a specific inhibitor of β-glucosidase, dramatically inhibited enzyme activity. When the concentrations were 10 and 1 mM, the BcXyl activity was only 5.7% ± 0.8 and 13.2% ± 0.69 compared to the control. In addition, 10 mM silver and mercury ions also showed significant inhibition of BcXyl, with 8.1% ± 0.63 and 6.5% ± 0.97 respectively relevant to the control activity. Other ions, such as 10 mM Mg^2+^, Cu^2+^, Mn^2+^, and Ca^2+^ showed no inhibition of BcXyl (Table 4).

### 2.5. BcXyl Showed Glycosidase Activity to the Highly Purified Anthocyanins from B. calycina

Two main anthocyanin peaks were purified by Sephadex LH20 column chromatography. The first peak was collected and combined as purified anthocyanin substrate I, from tube No. 70 to 77, with absorbance at 510 nm from 0.4 to 0.55 (Figure 4A). The second peak was collected and combined as purified anthocyanin substrate II, from tube No.120 to 138, absorbance at 510 nm was from 0.6 to 0.95 (Figure 4A). Anthocyanin degradation by the above purified BcXyl was analyzed by HPLC (high performance liquid chromatography). Substrate I was prepared at pH 7.0 and 2 major anthocyanin components were detected at 0 h, with retention time at 17.174 (P1) and 17.227 (P2) min (Figure 4B). After 2 h reaction at 40 °C,71% of P1 was degraded when incubated with BcXyl, while 49% of P1 degraded was with denatured BcXyl (Figure 4E), indicating that the native BcXyl led to significantly more anthocyanin degradation. No difference in the decrease of P2 content was found between the incubation with the native and denatured BcXyl, indicating non-enzymatic anthocyanin degradation occurred during the incubation (Figure 4C–E). No obvious decrease in anthocyanin content was detected when the substrate II was incubated with both native or denatured BcXyl, indicating the anthocyanins in substrate II were more stable and more resistant to BcXyl hydrolysis than those in substrate I.

To further confirm the ability of BcXyl to hydrolyze the glycosidic bond of anthocyanins, the release of sugars after incubation of the enzyme with anthocyanin substrate I was monitored by GC/MS (gas chromatography-mass spectrometry). When the anthocyanin substrate I was incubated with denatured BcXyl, no xylose derivatives were detected in the reaction mix (Figure 5B,C). However, when the native enzyme was added, xylose derivative (d-Xylononitrile, 2, 3, 4, 5-tetraacetate) of 1.05 ± 0.0518 μM was detected after 2 h of incubation (Figure 5A,C, standard curve sees Appendix A). In addition, galactose derivative (d-(+)-Galactose, aldononitrile, pentaacetate) was detected after 2 h-incubation with native or denatured BcXyl. However, higher levels of galactose derivatives were found for the reaction with native BcXyl (10.0 ± 0.6465 μM) than that with denatured BcXyl (2.51 ± 0.2652 μM) (Figure 5C, standard curve, see Appendix A).

### 2.6. Full-length BcXyl mRNA Sequence and Its Abundance during Flower Development

Based on the peptide sequences of the BcXyl (Table 2), degenerate primers (Appendix A) were designed to amplify the cDNA fragments encoding the enzyme. A full-length mRNA sequence (NCBI No. MK411219) of 2358 bp including the complete coding sequence (CDS) and 3′and 5′un-translate region (UTR) were achieved (Appendix A). We performed qPCR analysis of the gene expression of *BcXyl* at the four flower development stages of *B. calycina*. The expression of *BcPrx01*, coding for a basic peroxidase and known to play an important role in the anthocyanin degradation in the flowers (*B. calycina*) [12], was also analyzed. The relative gene expression level of *BcXyl* was low at stage 1 and 2, and rapidly increased at stage 3, to 140-fold higher expression than at stage 1, and then the level decreased at the fourth stage (Figure 6A). *BcXyl* showed similar expression patterns as *BcPrx01* (Figure 6A), implying that BcXyl may play a role in the rapid anthocyanin degradation of *B. calycina*. 

The CDS of *BcXyl* encoded a protein of 786 amino acids (Figure 6B) and a molecular mass of around 72 kDa (Figure 2C). N- and C-terminus of the members of glycoside hydrolase family 3 (GH3) and the fibronectin type III domain (this domain is frequently found in GH3, but its function is unknown) were detected in the sequence. A signal peptide which included the first 27 amino acids of BcXyl was detected at the N-terminus by SignalP4.1 (http://www.cbs.dtu.dk/services/SignalP/), and was predicted as the signal peptide for a ‘secretion protein’. The destination of the secretion of the protein might be either vacuole or extracellular space via the endoplasmic reticulum (ER) (Figure 6B). 

A phylogenetic tree of the amino acid sequences of glycoside hydrolase family 3 (GH3) was constructed by the neighbor-joining method. According to Cournoyer and Faure (2003) [21], two clads were observed in the phylogenetic tree, BcXyl was included in the subfamily AB of GH3 that included beta-glucosidase of *Kuraishia capsulata* (Kc-AAA91297; Bgln) and beta-glucosidase of *Bjerkandera adusta* (Ba-AUW34340; BadGluc) (Figure 6C). The two fungi enzymes had been characterized and were found to degrade anthocyanins [19,22]. In the alignment of BcXyl with Bgln, BadGluc and *Arabidopsis* Xylosidases (AtXyl 1 and 4), we found that the catalytic nucleophile Asp (black triangle) and four amino acid residues which might be involved in substrate binding (red triangle), were conserved in these AB clad members of glycoside hydrolase family 3 (GH3) [22] (Figure 6D).

## 3. Discussion

Anthocyanins are important plant pigments, yet very little is known about their degradation mechanism in planta [4]. Rapid anthocyanin degradation occurs during flower development of *Brunfelsia calycina*. In this study, our goal was to find out whether β-glycosidases were functioning in anthocyanin degradation. We isolated a β-xylosidase (BcXyl) from the petals of *B. calycina* and found the enzyme showed activity to anthocyanins.

### 3.1. A β-Xylosidase was Purified from B. calycina Flower and Proved to Catalyze the Hydrolysis of the Glycosidic Bonds in Anthocyanins

β-glucosidases that are able to hydrolyze the glycosidic bonds of anthocyanins, have long been found in microorganisms, so called anthocyanin-β-glycosidase or anthocyanase [23]. The released aglycones from anthocyanins by the β-glucosidases are susceptible to degradation [4]. *Bgln*, a gene coding for a β-glucosidase/anthocyanase from *Kuraishia capsulata* was isolated and transformed to *Saccharomyces cerevisiae*. The transformed yeast successfully produced brandy wines from red grapes, demonstrating the anthocyanin degradation function of the gene [19,24,25]. Anthocyanin reduction was found in the fruit juice of Sicilian blood orange during fruit ripening. β-glucosidase (BG) activities in the fruit juice showed an 80% correlation coefficient with anthocyanin concentration during fruit ripening, indicating that BG plays a role in anthocyanins degradation at the end of fruit ripening. However, the specific BG protein has not been identified by protein sequencing and gene coding as the enzyme has not been isolated [10].

Like anthocyanins, many secondary metabolites in plants exist as stable glycosylated compounds. The biological function of these secondary metabolites relies on the release of aglycones by hydrolysis of glycosidic bonds. To date, some flavonoid glycosidases have been purified and functionally characterized [26,27,28]. For example, a glucosidase (GmICHG) isolated from soybean root can hydrolyze genistein 7-*O*-(6″-*O*-malonyl-*β*-D-glucoside) to produce free genistein, which acts as a chemical attractant and plays a very important role in defense against pathogen infection [13]. Bar-Akiva (2010) showed that during the process of rapid anthocyanin degradation in the petals of *B. calycina*, 19 volatile aroma substances containing phenol rings were detected [20], implying the possibility of the presence of β-glycosidases to hydrolyze glycosylated phenolic volatile compounds.

In the present study, a β-glycosidase was purified from the petals of *B. calycina* to homogenous and was identified to be a β-xylosidase (BcXyl) by protein sequencing. Incubation of BcXyl with highly purified *B. calycina* anthocyanin led to significant decrease in the pigment concentration (Figure 4). In addition, we measured the release of hexoses by BcXyl via GC/MS monosaccharide detection, and found that xylose and galactose were released from the anthocyanins (Figure 5). The results proved that BcXyl is a β-xylosidase that is able to catalyze the hydrolysis of the xylosidic and galactosidic bonds of anthocyanins. To the best of our knowledge, this is the first anthocyanin-β-glycosidase identified in higher plants.

### 3.2. BcXyl Belongs to Glycoside Hydrolase Family 3 as Two Fungal Characterized Anthocyanin-β-Glycosidases

So far, four beta-glycosidases (anthocyanases) were reported to possess the ability to degrade anthocyanins. Of these four, two enzymes were from plants, including a probable β-glucosidase from eggplant, which decolorized anthocyanins [23] and from Sicilian blood orange [10], and two were from fungi, a β-glucosidase from *Aspergillus niger* [18] and a β-glucosidase (Bgln) from *Kuraishia capsulata* [19]. Among these four glucosidases, only *K. capsulate* Bgln was sequenced and the gene coding for the enzyme has been functionally characterized to be of the capability to degrade anthocyanin by heterogenous expression analysis [19]. Recently, a β-glucosidase (BadGluc) from *Bjerkandera adusta* was found to have anthocyanase properties. Both Bgln and BadGluc belong to the AB subfamily of glycoside hydrolase family 3 (GH3), which includes β-glucosidases (EC 3.2.1.21) and β-xylosidases (EC 3.2.1.37) [21,22]. In the present study, BcXyl is also a member of the AB subfamily of GH3 (AB-GH3), and shares 30.87% and 34.32% similarities to Bgln and BadGluc amino acid sequences respectively, and shared the conserved catalytic Asp and the putative substrate binding domains (Figure 6C,D). The results indicate that the certain conserved structure of the AB-GH3 members, either from higher plants or microorganism, may be required for the hydrolysis anthocyanin glyosidic bond. 

The optimum pH for the reactions catalyzed by BcXyl and BadGluc was around pH 5.0. BcXyl showed activity to both pNPGa and pNPX, while exhibiting no activity on pNPG (Table 3). BadGluc showed activity to both glucosidic and galactosidic anthocyanins, and Bgln showed activity to both pNPG and pNPX. The data indicate the anthocyanin-β-glycosidases from either fungi or higher plants may be of similar enzymatic features.

It is worth noting that we detected the activity of BcXyl on *B. calycina* anthocyanins at pH 7.0 (Figure 4 and Figure 5), not at the optimum pH (5.0) for the enzyme on pNPGa (Figure 3B). It is known the color, stability and molecular structure of anthocyanins are highly dependent on acidic conditions. Anthocyanin molecules exist as flavylium cations at low pHs, and present as pseudobases at pH 4.0–5.0, while as chalcones at pH 6.0–7.0 [29]. The change in molecular structure or polarity may influence the binding of the substrate for BcXyl. We estimate that BcXyl may only act on the chalcone type anthocyanins at pH 7.0. Under this condition, the pigments are unstable and non-enzymatic degradation may occur to interfere the activity assay. Actually, we did see the interference in the activity assay by HPLC (Figure 4) and by GC/MS (Figure 5). To rule out the interference, we set up denatured enzyme control in parallel. Under the assay condition, we detected that the pigment degradation or release of sugar by native enzyme was significantly more than those by denatured enzyme, indicating the activity of BcXyl on the anthocyanins.

### 3.3. The Role of BcXyl in the Rapid Degradation of Petal Anthocyanins

Vaknin (2005) found that increase of peroxidase activity was well correlated in time with the rate of anthocyanin degradation in *B. calycina* petal [5]. The peroxidase was identified as BcPrx01, and its activity was demonstrated in vacuole and showed anthocyanin degradation in vitro [12]. In this study, we compared the expression levels of the *BcXyl* and the *BcPrx01* in *B. calycina* petal development. Both genes showed dramatic increase in expression levels at stage 3, which corresponded to the rapid anthocyanin degradation in the petals (Figure 6A). In addition, we analyzed the BcXyl protein sequence and found secretion pathway (SP) signal peptide at N-terminusm as *BcPrx01*, which may be translocated to vacuole via ER pathway (Figure 6C). Accordingly, we speculate that in the vacuoles of *B. calycina* petals, BcXyl may first hydrolyze the glycosidic linkages of anthocyanins and release anthocyanidin, which facilitates the further degradation by BcPrx0l. However, BcXyl showed high specificity to the xylosidic and galactosidic anthocyanins, which may limit its ability to catalyze the degradation of other anthocyanins. High substrate specificity was also found for some flavonoid glycosidases. The soybean GmICHG showed high activity to isoflavone glycosides, while displaying negligible activity to anthocyanins [13].

In conclusion, a β-xylosidase (BcXyl) was isolated from *B. calycina* petal, and was found to possess the activity to hydrolyze the glycosidic linkages of anthocyanins. BcXyl belongs to the same AB subfamily of GH3 as two characterized fungal anthocyanin-β-glycosidases. BcXyl gene was also cloned and showed co-expression patterns as an anthocyanin degradation peroxidase gene (*BcPrx01*).

## 4. Materials and Methods

### 4.1. Plant Materials

*Brunfelsia calycina* (Cham. & Schltdl.) Benth. petals at four stages (flower bud, deep purple, light purple and white petals, indicated as stage 1, 2, 3 and 4 in Figure 1A) were collected respectively in the campus of South China Agricultural University, Guangzhou City, the South-East of China (23°7′ N, 113° E). The sampled petals were immediately analyzed or were quickly frozen in liquid nitrogen and stored at −80 °C until use.

### 4.2. Anthocyanin Content Determination

Anthocyanin content determination was performed as described by Luo (2017) [30], using 0.15 M aqueous HCl solution for anthocyanin extraction and a pH-differential method for content determination.

### 4.3. Crude Enzyme Extraction, protein content and β-Glycosidase Activity Assay

Crude enzymes of the petals at each stage were respectively extracted by grinding 2 g of the petal samples in liquid N_2_ and then homogenizing the sample powder with 8 mL of 0.1 M potassium phosphate buffer (KPB; pH 7.0), containing 8.6 mM dithiothreitol (DTT), 5 mM ethylenediaminetetraacetic acid (EDTA), 1 mM phenylmethylsulfonyl fluoride (PMSF) and 5% (*w/v*) PVPP. After centrifugation at 12,000× *g* and 4 °C for 20 min, the supernatants were collected as crude enzyme extracts [13]. Protein content of the extracts was measured by using Bradford protein assay kit (Sangon Biotech, Shanghai, China) according to the manufacturer’s protocol. Bovine serum albumin was employed as a protein standard.

β-Glycosidase activity assay was performed according to Li (2018) [31] with minor modifications. The reaction mixture included 50 μL of the crude enzyme extracts, as well as 200 μL of 10 mM p-Nitrophenyl β-D-galactopyranoside (pNPGa, Sigma-Aldrich, Saint Louis, MO, USA) in McIlvaine buffer (Citrate Phosphate Buffer, 0.1 M, pH 5.0). The reaction mixture was incubated at 50 °C for 30 min, and the reaction was terminated by adding 2 mL of 1 M Na_2_CO_3_. The denatured reaction was set up in parallel with enzyme that had been boiled for 10 min. The absorbance was measured at 405 nm (p-nitrophenol extinction coefficient = 18300 M^−1^cm^−1^) [19]. One unit of the β-glycosidase activity is defined as the amount of enzyme required to release 1 μmol of pNPGa per hour under the above assay conditions.

### 4.4. Purification and Identification of a β-Xylosidase in B. calycina Petals

The crude enzyme extraction was performed as described above with 50 g of stage 2 petal in 250 mL of the extraction solution. The crude extracts were first fractionated by precipitation with ammonium sulfate at 25% to 75% saturation and centrifugation at 12,000× *g* for 20 min. The pellet was dissolved in 0.1 M KPB (pH 7.0) containing 8.6 mM DTT, 1 mM EDTA, 0.2 mM PMSF and dialyzed against 10 mM KPB (pH 7.0) containing 1.1 mM DTT (buffer A).

According to Suzuki (2006) [13] with minor modifications, the dialyzed enzyme solution was applied to a DEAE-Sepharose column (1.5 cm × 50 cm, GE Healthcare Bio-Sciences AB, Uppsala, Sweden) previously equilibrated with buffer A. β-glycosidase was eluted from the column at a flow rate of 0.3 mL/min with buffer A. Fractions of 2 mL were collected for both β-glycosidase activity assay (see “4.3”) and protein concentration measurement (A280 nm). The fractions with high β-glycosidase activity were combined and concentrated by an ultrafiltration centrifuge tube (Amicon Ultra-15 Centrifugal Filter Units, Merck KGaA, Darmstadt, Germany). The concentrated fraction was further loaded onto a CM-cellulose column (1.5 cm × 50 cm, Sigma-Aldrich, Saint Louis, MO, USA) previously equilibrated with buffer A. The column was first washed with buffer A and then was eluted with an NaCl linear gradient (0–1 M) in buffer A at a flow rate of 0.2 mL/ min. Fractions of 1 mL were collected and assayed as described above. The fractions with the highest specific activity (ratio of activity to protein) were pooled for the purity analysis.

The purified protein was separated in 10% (*w/v*) sodium dodecyl sulfate-polyacrylamide gel electrophoresis (SDS-PAGE) after 10 min of boiling in Laemmli’s sample buffer following standard conditions [32]. The gel was stained with Coomassie Brilliant Blue R-250 (Sigma-Aldrich) to check the purity of the proteins. The band of 72 kDa in the gel was excised and sent for protein sequencing by Sangon Biotech (Shanghai, China) via Biolynx peptide sequencing, peptide sequence search was carried out by MS/MS ion search using MASCOT version 2.5.0 (Matrix Science, London, UK). The raw MS/MS data were searched against NCBI protein sequences, which were updated on August 11, 2017 (NIH, Bethesda, MD, USA). Viridiplantae was used as taxonomy.

### 4.5. Effect of pH and Temperature on Enzyme Activity

To determine the optimal temperature, the enzyme activity of the purified β-glycosidase, nominated as β-Xylosidase (BcXyl) hereafter based on the peptide sequencing, was measured at different temperatures ranging from 30 °C to 70 °C using the activity assay method as described in “4.3”. For optimal pH determination, 10 mM pNPGa was prepared in a set of 0.1 M McIlvaine buffers of pH 3.0 to pH 7.0. Enzymatic assays were performed as described in “4.3”.

### 4.6. Influence of Metal Ion and Chemical Effectors on Enzyme Activity

BcXyl activity was assayed as described in “4.3” containing 10 mM pNPGa with 1 or 10 mM effector solutions (AgNO_3_, HgCl_2_, MgCl_2_, CuSO_4_, MnCl_2_, CaCl_2_, d-gluconic acid). The relative activity was defined as the relative value to the activity of control without the effectors.

### 4.7. Substrate Specificity

Substrate specificity for different β-glycosidic p-nitrophenyl compounds was tested with the purified BcXyl. Reactions were carried out in 0.1 M McIlvaine buffer (pH 5.0) at 50 °C as described in “4.3”. The compounds tested were: 10 mM of p-nitrophenyl-β-d-galactopyranoside (pNPGa), p-nitrophenyl-β-d-xylopyranoside (pNPX), p-nitrophenyl-β-d-mannopyranoside (pNPM), p-nitrophenyl-β-d-glucopyranoside (pNPG).

### 4.8. Anthocyanin Purification

*B. calycina* petals (100 g) were blanched with 1 L of 0.3 M aqueous HCl and the red extracts were collected and filtered through qualitative filter paper (Grade 1, medium flow, Whatman^®^, Sigma-Aldrich). The filtrate was then subjected to Amberlite XAD-7 resin (Sigma-Aldrich) column (2.5 × 30 cm) chromatography according to Zhang (2004) [33]. Fractions with the highest A510 nm were pooled and concentrated by removing the methanol at 50 °C in a rotary evaporator (Heidolph, Schwabach, Germany). The concentrated anthocyanins were then loaded onto a Sephadex LH-20 column (1.0 × 90 cm, Sigma-Aldrich), eluted with 10% (*v/v*) aqueous FA (formic acid) at a flow rate of 0.3 mL/ min. Fractions (3 mL/tube) were collected and monitored by the absorbance at 510 nm. The fractions in the major peaks of the A510 nm elution profile were pooled and subjected to Amberlite XAD-7 resin to remove the residual FA and then the elution was concentrated as purified anthocyanins. The purified anthocyanins were filtered through the 0.22 μm PVDF membrane and transferred to a vial before analysis.

### 4.9. High Performance Liquid Chromatography (HPLC) Analysis of Anthocyanin Degradation by Brunfelsia β-Xylosidase (BcXyl)

The above purified anthocyanins were concentrated and prepared in McIlvaine buffer (0.1 M, pH 7.0), as degradation reaction substrates. 200 μL of the substrate was incubated with the purified BcXyl (50 μL) for 2 h and the reaction was terminated by the addition of 200 μL of a chloroform-methanol (1:1) solution containing 0.2% aqueous HCl. Control reaction was set up in parallel with the boiled enzyme. After the reaction mixture was centrifuged at 12,500× *g* for 5 min, the upper aqueous phase was collected and analyzed by HPLC on an equipped Agilent 1200 Series HPLC system (Agilent Tech., Santa Clara, CA). The upper phase of 20 μL was injected into a C18(2) column (Luna^®^, 5 mm, 250 ± 4.6 mm; Phenomenex, Torrance, CA). The mobile phase consisted of 0.1% formic acid (FA) in acetonitrile (A) and 0.1% FA in water (B). Gradient elution at a flow rate of 0.8 mL/min was used from 10% to 35% A over 35 min at 35 °C. The monitoring wavelength was 280 nm and 510 nm.

### 4.10. Identification and Quantification of Sugars Released from Anthocyanin Degradation by BcXyl by Gas Chromatography-Mass Spectrometry

For GC-MS analysis of the sugars released from anthocyanins, the reaction was performed as described in “4.9” but terminated by chloroform. After centrifugation, the upper aqueous phase was freeze-dried overnight. Then 0.5 mL of pyridine, 10 mg of hydroxylamine hydrochloride were added to the dried sample, and the mixture was incubated at 90 °C for 30 min. After cooling to room temperature, 1 mL of acetic anhydride was added, and the mixture was incubated again at 90 °C for 30 min. After cooling, the above derivatized mixture was filtered through 0.22 μm PVDF membrane before GC-MS analysis. The sample was injected into a gas chromatography/mass spectrometry system (GCMS-QP2010 Ultra, Shimadzu, Japan). A HP-5 MS capillary column (30 m × 0.25 mm inner diameter, 0.25 μm film thickness) was used with helium as carrier gas, with 1.0 mL/min flow rate. Oven temperature was set at an initial temperature of 110 °C for 1 min, followed by an increase of 2 °C to 180 °C and a hold at 180 °C for 3 min, then at 10 °C min^−1^ to 220 °C and a hold at 220 °C for 3 min, followed by a ramp of 20 °C min^−1^ to 280 °C and a hold at 280 °C for 10 min. The injection port and the flame ionization detector were kept at 250 °C [34]. A quadruple mass detector with electron ionization at 70 eV was used to obtain the MS data in the range of 35–500 m/z. Identification of the sugars was done by matching the retention indices with those of the standard monosaccharides and their characteristic mass spectrum and by comparison of the MS spectral data with the NIST14 GC-MS library. Standard curves were plotted with the concentrations of the galactose and xylose standards against the peak areas obtained from the total ion chromatogram.

### 4.11. BcXyl cDNA Cloning, Sequence Features, Potential 3D Structure Prediction of the Deduced Protein Sequence

Total RNA was extracted by MiniBEST Plant RNA Extraction Kit (Takara, Japan). First-strand cDNA was synthesized using PrimeScriptTM RT Reagent Kit with gDNA Eraser (Takara). The cDNA was then used as a template to amplify the fragment of *BcXyl* with degenerate primers designed according to the peptide sequences (Appendix A). A 1275-bp fragment was amplified and used to design primers (Appendix A) for 3′ RACE and 5′ RACE to obtain the full-length cDNA sequence of *BcXyl* (Appendix A), using SMARTer^®^ RACE 5′/3′ Kit (Takara).

The deduced protein sequence encoded by a full-length cDNA sequence of the BcXyl was obtained by the open reading frame finder software in NCBI (http://www.ncbi.nlm.nih.gov). The putative signal peptide (N-terminal peptide) in the sequence was identified using the SIGNALP software (http://www.cbs.dtu.dk/services/SignalP/). The domain structures were inferred by BLAST software (http://blast.ncbi.nlm.nih.gov/Blast.cgi).

### 4.12. Molecular Phylogenetic Analysis of BcXyl

β-glycosidase sequences of the AB members of Glycoside hydrolase family 3 (AB-GH3) were collected based on Cournoyer and Faure (2003) [21]. Sequences from bacteria, fungi and plants are included. The deduced amino acid sequence of *BcXyl* were aligned with these AB-GH3 sequences by MEGA7 software [35]. A rooted neighbor-joining phylogenetic tree was constructed by the MEGA7 software.

### 4.13. BcXyl Transcript Abundance Analysis

qRT-PCR was applied to analyze the gene transcript abundance of *BcXyl* and *BcPrx01* during flower development. cDNA was prepared from 1 μg of total RNA with PrimeScript reverse transcriptase (Takara). Specific primers for the genes were designed and are described in Appendix A. *B. calycina 18s rRNA* was selected as a reference gene. qPCR was performed in a total volume of 20 μL, containing 100 nM each primer and 10 μL of SYBR Green PCR Supermix (Toyobo) on a Bio-Rad CFX96 Real-Time PCR system (Bio-Rad, Hercules, CA, USA).

### 4.14. Statistics Analysis

Statistics data was collected from three biological replicate samples. Values were presented as means ± standard error of mean (SEM). Means were compared by unpaired t test (*p* < 0.05) using the GraphPad QuickCalcs online software (Prism 7, GraphPad Software, San Diego, CA, USA).

Sequence data used in this article can be found in the GenBank data libraries (http://www.ncbi.nlm.nih.gov) under the following accession numbers: *Brunfelsia calycina*, MK411219 (*BcXyl*); *Kuraishia capsulata*, AAA91297 (*Bgln*); *Bjerkandera adusta*, AUW34340 (*BadGluc*); *Arabidopsis thaliana*, Q9FGY1, Q9FLG1, BAB09906, BAB11424, Q9LXD6, Q9LXA8; *Trichoderma reesei*, Q92458; *Aspergillus niger*, O00089; *Aspergillus nidulans*, O42810; *Aspergillus oryzae*, O42698; *Xylella fastidiosa*, AAF83655; *Bacillus halodurans*, BAB05627; *Streptomyces coelicolor*, CAB91121; *Pseudomonas aeruginosa*, AAG05115; *Salmonella typhimurium*, Q56078; *Escherichia coli*, AAG57264, P33363; *Flavobacterium meningosepticum*, O30713; *Bacteroides fragilis*, O31356.

## Figures and Tables

**Figure 1 ijms-20-01423-f001:**
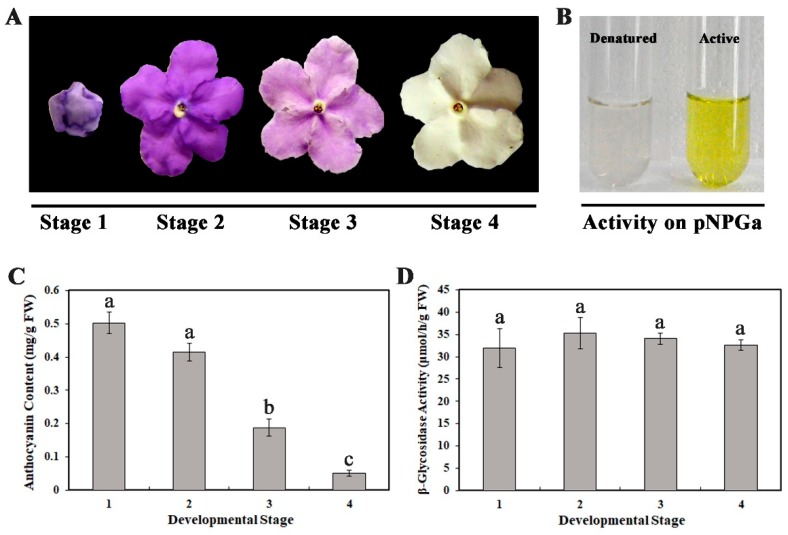
Anthocyanin degradation and β-glycosidase activity during flower development of *Brunfelsia calycina*. (**A**) The images of the flowers at different developmental stages. Stage 1 to 4 indicates purple buds, dark purple, light purple and white flower petals respectively. (**B**) β-glycosidase activity in the crude enzyme extract from *B. calycina*. pNPGa (p-Nitrophenyl β-D-galactopyranoside) turned yellow due to the β-glycosidase activity, with the boiled extract (denatured) as a reference. (**C**) Change in anthocyanin contents during flower development. (**D**) β-glycosidase activity during flower development. The values presented in C and D are means of three measurements from three individual extractions. Error bars indicate the standard error of mean (SEM) of the values. Different letters denote significant differences in the values according to unpaired t test (*p* < 0.05), while same letters denote non-significant differences in the values.

**Figure 2 ijms-20-01423-f002:**
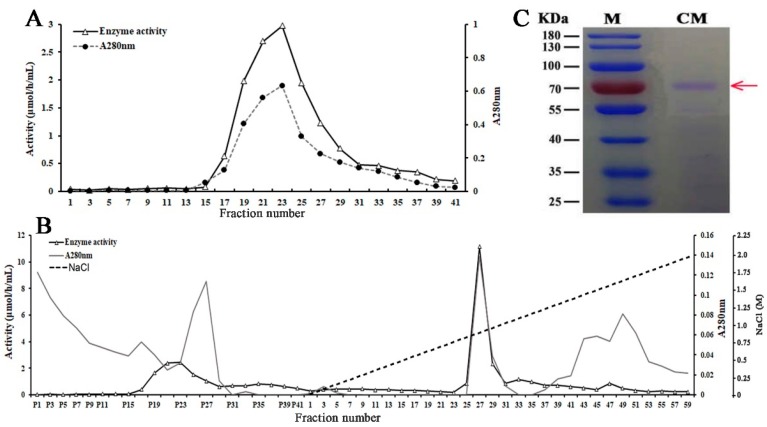
Purification of β-glycosidases from *B. calycina* petals. (**A**) Total protein (A280 nm) and β-glycosidase activity determined in DEAE (Diethylaminoethyl)-Sepharose column chromatography fractions from petal crude enzyme extract. The major activity fractions were collected for further purification. (**B**) Total protein (280 nm) and β-glycosidase activity determined in CM (Carboxymethyl)-Sepharose column chromatography fractions. The fractions collected as indicated in (A) was further separated by CM-Sepharose column chromatography and the fractions in an activity peak that overlapped with a protein peak (Fraction No. 26–28) were collected for purity analysis. Fractions from P1 to P41 were the elution by 10 mM potassium phosphate buffer (KPB, pH 7.0) containing 1.1 mM DTT (Dithiothreitol). Fraction 1 to 59 were eluted with an NaCl linear gradient (0–1 M) in the above described KPB buffer. (**C**) Urea-SDS-PAGE (urea sodium dodecyl sulfate polyacrylamide gel electrophoresis) of the purified glycosidase. Fractions as indicated in (B) were subjected to Urea-SDS-PAGE and major band that for sequencing was marked by red arrow.

**Figure 3 ijms-20-01423-f003:**
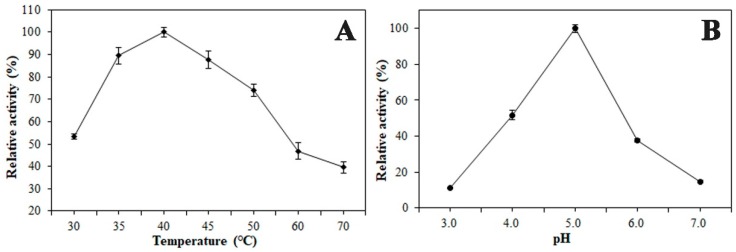
Temperature and pH optima of the *B. calycina* β-glycosidase /β-xylosidase (BcXyl). (**A**) Temperature optima of BcXyl on pNPGa. (**B**) pH optima of BcXyl on pNPGa. Each point represents the average of three experiments, ± standard error.

**Figure 4 ijms-20-01423-f004:**
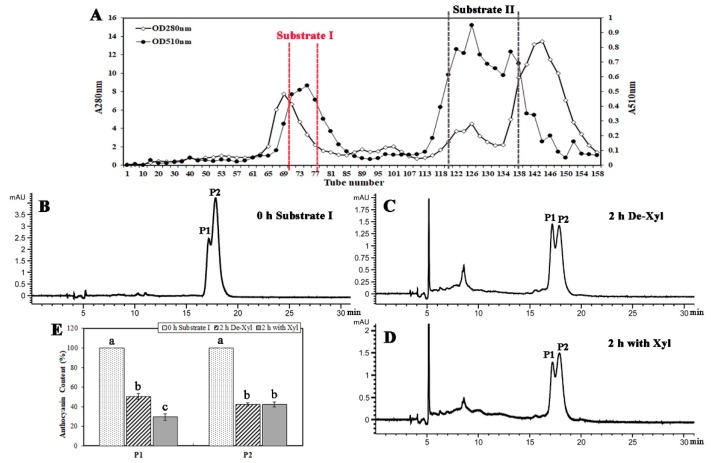
Anthocyanin degradation activity on purified *B. calycina* anthocyanins by BcXyl. (**A**) *B. calycina* anthocyanin purification profiles by column chromatography. Absorbance at 510 nm and 280 nm determined in the fractions of Sephadex LH-20 column chromatography of the anthocyanins from *B. calycina* flowers. The fractions in the anthocyanin peaks as indicated between two dotted lines were collected as substrate I and II for the anthocyanin degradation activity assay. (**B**–**D**) HPLC (high performance liquid chromatography) analysis of *B. calycina* anthocyanin after incubation with BcXyl. HPLC profiles of the anthocyanin substrate I as indicated in (A) before incubation with BcXyl (B), after 2 h-incubation with native (D) and denatured BcXyl (C), are shown. Two major anthocyanins (P1 and P2) was detected in the substrate I. (**E**) Relative contents of anthocyanin P1 and P2 after incubation with BcXyl. The incubation reactions are as described in (B–D), and relative anthocyanin contents are represented by the peak area in the HPLC profiles. The statistical details of the values presented in (E) are as described in Figure 1.

**Figure 5 ijms-20-01423-f005:**
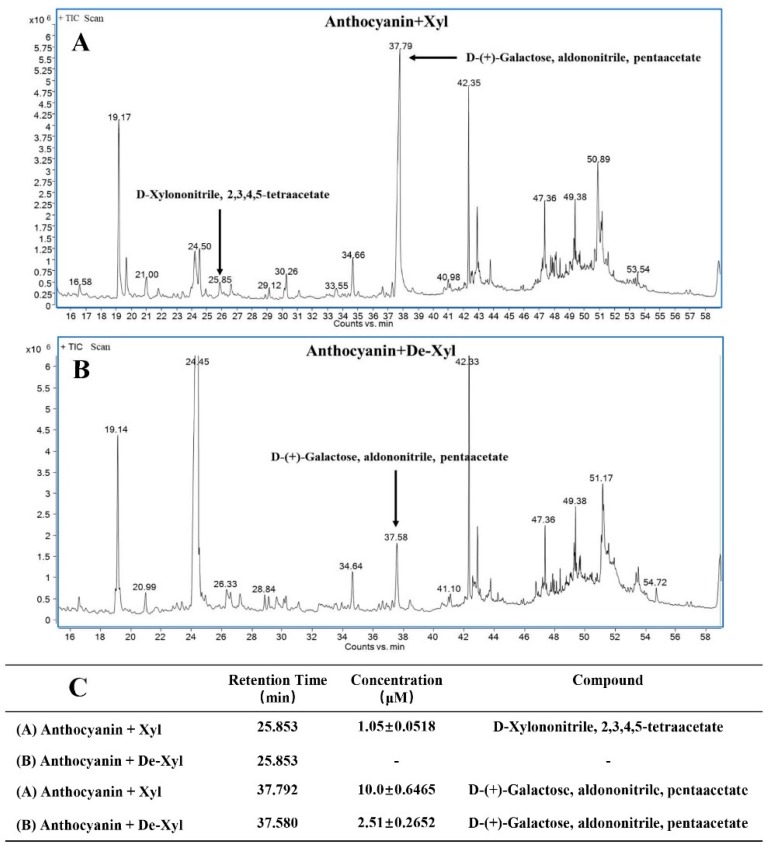
GC-MS detection of the monosaccharides released from the *B. calycina* anthocyanins by the action of BcXyl. (**A,B**) GC-MS (gas chromatography-mass spectrometry) chromatograms for monosaccharide detection after anthocyanin degradation by BcXyl. The degradation reactions were set up as described in (Figure 4B–D). The release of monosaccharides from 6.4 µM anthocyanins incubated with native (A) and denatured (B) BcXyl for 2 h were subjected to GC/MS analysis. (**C**) The contents of the detected monosaccharides in (A,B). Concentrations of monosaccharides were based on the comparison of the relevant peak area in the GC-MS chromatograms to the standard curves. The values are means ± standard deviations (*n* = 3).

**Figure 6 ijms-20-01423-f006:**
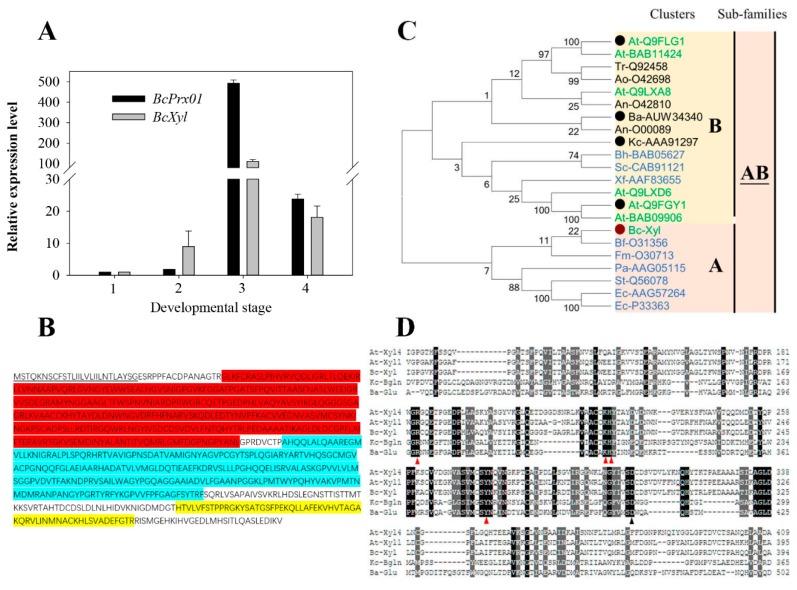
Gene expression and full-length amino acid sequence analysis of a BcXyl. (**A**) Relative expression levels of *BcXyl* and *BcPrx01* during the flower development of *B. Calycina*. The relative levels of the genes were acquired by qPCR with *B. calycina 18s rRNA* (L49274) as internal reference. (**B**) Domains in the deduced amino acid sequence of *BcXyl*. Putative domains include: a secretion protein (SP) signal peptide (doubly underlined); N-terminal domain of Glycoside hydrolase family 3 (GH3) (red); C-terminal domain of GH3 (light blue); fibronectin type III domain of GH3 (yellow). (**C**) A rooted neighbor-joining phylogenetic tree of the amino acid sequences of subfamily AB from GH3. The sequences from bacteria, fungi and plants, indicated by blue, black and green respectively. Two clusters (A and B) are based on clusters classification of GH3 by Cournoyer and Faure (2003) [21]. The red dot indicates BcXyl and the black dots high light sequences included in the alignment in (D). (**D**) Amino acid sequence alignment analysis of BcXyl with four members from cluster AB of GH3. Multiple sequence alignment using COBALT including AtXyl1 (Q9FGY1) and AtXyl4 (Q9FLG1) from *Arabidopsis thaliana*; Bgln from *Kuraishia capsulata* (AAA91297) and BadGluc from *Bjerkandera adusta* (AUW34340). Conserved areas are shadowed in grey/black, the catalytic nucleophile Asp is marked with a black triangle and red triangles represent the predicted amino acid residues, which might be involved in substrate binding.

**Table 1 ijms-20-01423-t001:** Summary of the Purification Process of β-Glycosidases from *B. calycina* petals.

Purification Step	Total Activity (μmol/h) ^1^	Total Protein (mg)	Specific Activity (μmol/h/mg)	Purification (fold)	Yield (%)
crude extract	614.64	85.93	7.15	1.00	100.00
(NH_4_)_2_SO_4_	295.74	13.96	21.19	2.96	48.12
DEAE Sepharose ^2^	136.89	3.68	37.23	5.21	22.27
CM-Sepharose ^3^	22.30	0.07	318.57	44.56	3.63

^1^ One unit of β-glycosidase activity is defined as the amount of enzyme that catalyzes the release of 1 μmol of p-nitrophenol from pNPGa per hour under the assay conditions. ^2^ DEAE = Diethylaminoethyl. ^3^ CM = Carboxymethyl.

**Table 2 ijms-20-01423-t002:** Fragment sequences of the *B. calycina* β-glycosidase detected by tandem mass spectrometry (MS/MS) and their identical hits from other plant species.

	Sequences Detected by MS/MS	Annotation	Gene Identified	Organisms
1	AVSNNFATLMR	beta-xylosidase/alpha-l-arabinofuranosidase	XP_009612011.1; XP_006351808.1; XP_009762535.1	*Nicotiana tomentosiformis; Solanum tuberosum*; *Nicotiana sylvestris*
2	LPMTWYPQSYADK	beta-xylosidase /alpha-l-arabinofuranosidase	XP_009612011.1; XP_009762535.1	*Nicotiana tomentosiformis*; *Nicotiana sylvestris*
3	VTQQDLDDTFNPPFK	beta-xylosidase /alpha-l-arabinofuranosidase	XP_009612011.1; XP_006351808.1; XP_009762535.1	*Nicotiana tomentosiformis; Solanum tuberosum; Nicotiana sylvestris*
4	HYTAYDIDDWK	beta-xylosidase/alpha-l-arabinofuranosidase	XP_006351808.1	*Solanum tuberosum*
5	YEWWSEALHGISYTGPGVK	beta-xylosidase/alpha-l-arabinofuranosidase	XP_006351808.1; XP_009762535.1	*Solanum tuberosum*; *Nicotiana sylvestris*

**Table 3 ijms-20-01423-t003:** Substrate specificity of BcXyl.

Glycoside Substrate	Glycosidase Activity (μmol/(h·mg Protein)	Relative Activity (%)
p-Nitrophenyl β-d-galactopyranoside	301.35 ± 4.1	100 ± 1.6
p-Nitrophenyl β-d-xylopyranoside	118.62 ± 2.4	39.36 ± 0.7
p-Nitrophenyl β-d-mannopyranoside	0	0
p-Nitrophenyl β-d-glucopyranoside	0	0

The activity with p-Nitrophenyl β-d-galactopyranoside (pNPGa) was taken as 100% activity. Each value represents the average of three experiments, ± standard error.

**Table 4 ijms-20-01423-t004:** Effect of different substances on the activity of BcXyl.

Substance	Concentration (mM)	Residual Activity (%)
d-gluconic acid	10	5.7 ± 0.8
	1	13.2 ± 0.69
AgNO_3_	10	8.1 ± 0.63
	1	11.9 ± 0.6
HgCl_2_	10	6.5 ± 0.97
	1	59.6 ± 0.44
MgCl_2_	10	98.9 ± 3.0
	1	100.1 ± 3.4
CuSO_4_	10	95.2 ± 1.5
	1	100.5 ± 2.0
MnCl_4_	10	103.5 ± 4.2
	1	100.6 ± 2.1
CaCl_2_	10	100.5 ± 2.2
	1	102.9 ± 1.5
Control	—	100 ± 1.0

The activity with pNPGa as indicated in Table 3 was taken as control. Each activity value represents the average of three experiments, ± standard error.

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
