# Peer review of "BcXyl, a β-xylosidase Isolated from Brunfelsia Calycina Flowers with Anthocyanin-β-glycosidase Activity"

_ijms, 2019, doi:10.3390/ijms20061423_

Reviewer 1 Report

The manuscript is interesting for researchers in this field. Nevertheless, there are some incomprehensible points. A revision would contribute to a better quality. Line 112: Please explain the abbreviation FW (I have found the explanation in the paper of Luo et al., 2017) Line137/Line149: Tube numbers 26-28 are hardly detectable in the graphic, hence, it is difficult to see what is meant with “low” protein content since in the figure it looks like an increase of protein content with activity but maybe I am wrong? Line165-174: The assignment of identical proteins is not clear. First of all the amino acid sequences 2 and 4 in Table 2 are identical but are described to exhibit identities to different proteins. Why? Actually, the protein sequences XP_009612011 and XP_009762535 exhibit all 3 mentioned amino acid series. What is your assignment based on? The sequences deposited with the mentioned accession numbers were annotated as beta-xylosidase/alpha-L-arabinofuranosidase. How do the authors conclude that they may function as beta-xylosidase exclusively? Line183, Line186: I would not recommend to use the word “high” activity with 39 % or “dramatically” different activity to different substrates. A distinct substrate specificity is completely normal.
Line209: Was one reaction incubated without addition of enzyme/denatured enzyme to maybe confirm spontaneous anthocyanin degradation? Is it known that spontaneous degradation occurs? Or is it possible that there are other degrading components? Since the test was incubated for two hours, was the denatured enzyme tested for the same time period with pNPX to exclude that there might be residual activity over the time?
Figure 4: Since the y-axis in Figure 4C and 4D exhibit a different scaling, the peaks cannot be compared by the reader. Actually, it looks like the same level for the denatured enzyme when compared to the native enzyme? Line246: Is the NCBI number MK411219 confidential?
Line255: Specify the sentence “contains the N- and C-terminus of glycoside hydrolase family members”
Line339: The authors mention “negligible activity on pNPG”. Since in Fig. 3C the activity is stated “0”, I was expecting no activity at all. Different abbreviations were used for the nitrophenol-linked substrates. pNPG and pNPX are the more frequently used abbreviations (compared to ϱNP)

Author Response

Dear reviewer,

We are very appreciated for the critical and detailed comments,

here are our responses:

Point 1:

Line 112: Please explain the abbreviation FW (I have found the explanation in the paper of Luo et al., 2017)

Line137/Line149: Tube numbers 26-28 are hardly detectable in the graphic, hence, it is difficult to see what is meant with “low” protein content since in the figure it looks like an increase of protein content with activity but maybe I am wrong? 

Response 1: We have already written the full words for the abbreviation FW in the text for the first time. We modified Figure 2 to show the tube number more clearly and described more clearly in the legend.

Point 2: Line165-174: The assignment of identical proteins is not clear. First of all the amino acid sequences 2 and 4 in Table 2 are identical but are described to exhibit identities to different proteins. Why? Actually, the protein sequences XP_009612011 and XP_009762535 exhibit all 3 mentioned amino acid series. What is your assignment based on? The sequences deposited with the mentioned accession numbers were annotated as beta-xylosidase/alpha-L-arabinofuranosidase. How do the authors conclude that they may function as beta-xylosidase exclusively?

Response 2: Thanks for the comments. We are sorry that we did make mistake in Table 2. First, we should not put sequence 4 that is identical to sequence 2 in the table. We will delete sequence 4. You are right that XP_009612011 and XP_009762535 exhibit all the 3 amino sequences in the table.

Due to this mistake, we checked carefully the sequence reports again. We have run the protein sequencing for 3 times. One run was carried out after submission. The new results showed 2 additional sequences were detected. We add these additional 2 sequences to the table, and revise the table so as to let the readers to see that different peptides actually occur in the same protein hit.

You are right that all XP_009612011, XP_009762535 and XP_006351808 were predicted as beta-xylosidase/alpha-L-arabinofuranosidase using gene prediction method “Gnomon” (see in the NCBI information of XP_009612011). Based on the nomination of NCBI, we put the full name “beta-xylosidase/alpha-L-arabinofuranosidase “in the revised table.

All these proteins belong to Glycoside Hydrolase Family 3 (GH3). In this family, some members exhibit beta-xylosidase activity, some exhibit beta alpha-L-arabinofuranosidase activity, while some even exhibit both (Xiong et al., 2007). Our BcXyl showed high activity to p-Nitrophenyl β-D-galactopyranoside (pNPGa) and p-Nitrophenyl β-D-xylopyranoside (pNPX), and showed activity to the hydrolysis of the beta glycoside bond of anthocyanins. Accordingly, we assigned the BcXyl as beta-xylosidase. Whether BcXyl also shows alpha-L-arabinofuranosidase activity is required to be analysed in the future.

Xiong JS, Balland-Vanney M, Xie ZP, Schultze M, Kondorosi A, Kondorosi E, Staehelin C. 2007. Molecular cloning of a bifunctional beta-xylosidase/alpha-L-arabinosidase from alfalfa roots: heterologous expression in Medicago truncatula and substrate specificity of the purified enzyme. J Exp Bot. 58(11):2799-810.

Point 3: Line183, Line186: I would not recommend to use the word “high” activity with 39 % or “dramatically” different activity to different substrates. A distinct substrate specificity is completely normal. 

Response 3: Yes, we agree with the reviewer’s comment and already deleted the two words.

Point 4: Line209: Was one reaction incubated without addition of enzyme/denatured enzyme to maybe confirm spontaneous anthocyanin degradation? Is it known that spontaneous degradation occurs? Or is it possible that there are other degrading components? Since the test was incubated for two hours, was the denatured enzyme tested for the same time period with pNPX to exclude that there might be residual activity over the time? 

Response 4: we agree with the reviewer and find that “spontaneous” is not accurate, while “non-enzymatic” may be more accurate to describe the anthocyanin degradation without the function of enzyme. It is well known that isolated anthocyanins are highly instable and very susceptible to degradation even without any enzyme (Giusti & Wrolstad, 2003). Non-enzymatic degradation is known for anthocyanins in vitro, in particular when pH increases to 6-7 (Castañeda-Ovando et al., 2009), or temperature increases to above 50 (Peron et al., 2017). We are sure that the enzyme was completely denatured after boiling for 10 min (see figure 1B). We believe that since the reaction was at pH 7.0 and 40 , anthocyanins was instable at this condition and non-enzymatic degradation occurred.  This non-enzymatic degradation may be the so-called thermal degradation as suggested by Person et al., (2017), in which activation enthalpy is required for the thermal degradation.

Castañeda-Ovando, Araceli; Pacheco-Hernández, Ma. de Lourdes; Páez-Hernández, Ma. Elena. et al. (2009) Chemical studies of anthocyanins: A review. Food Chemistry, 2009, 113:859–871

Peron, D.V; Fraga, S; Antelo, F . (2017) Thermal degradation kinetics of anthocyanins extracted from juçara (Euterpe edulis Martius) and “Italia” grapes (Vitis vinifera). Food Chemistry, 232: 836–840

Point 5: Figure 4: Since the y-axis in Figure 4C and 4D exhibit a different scaling, the peaks cannot be compared by the reader. Actually, it looks like the same level for the denatured enzyme when compared to the native enzyme?

Response 5: We modified Figure 4 and made the y-axis in Figure 4C and 4D exhibit the same scaling. Based on the image of the HPLC profile, the difference between the denatured and native enzyme was not very obvious. When we used peak area to represent the relative content of anthocyanins, we detected significant difference of the decrease levels of P1 between the native and denatured enzyme (Figure 4E).  Since we were not so confident to the activity assay by HPLC,  we used GC-MS to further confirm our speculation in Figure 5.

Point 6:

Line246: Is the NCBI number MK411219 confidential?

Line255: Specify the sentence “contains the N- and C-terminus of glycoside hydrolase family members”

Response 6: We specified the sentencecontains the N- and C-terminus of glycoside hydrolase family members in the text. Else, the gene sequence has been uploaded to the NCBI website on Jan. 17th 2019. Since the paper has not been published, the uploaded gene sequence is temporarily kept secret. Screenshot of the email letter below can be used as a proof. We will ensure the NCBI number be public if our manuscript is accepted by the journal.                                 

Point 7: Line339: The authors mention “negligible activity on pNPG”. Since in Fig. 3C the activity is stated “0”, I was expecting no activity at all. Different abbreviations were used for the nitrophenol-linked substrates. pNPG and pNPX are the more frequently used abbreviations (compared to ϱNP)

Response 7: Yes, we agree with the reviewer and already corrected our careless mistakes.

Else, we also submit the responses via a word document, please check the attachment.

Best regards,

Xuequn Pang and Zhaoqi Zhang

Reviewer 2 Report

Abstract, L68

Please define the meaning of POD.

Figure 3B,Figure 4, L209, L442-445

Figure 3B suggest that relative BcXyl activity is highest in pH 5.0, and low in pH7.0.

However, in anthocyanin degradation activity assay in Figure 4, assay was performed at pH7.0. Why did authors perform the assay in this condition? In addition,  since “Spontaneous” anthocyanin degradation is high in this assay, it is recommended to consider assay condition.

Figure 3

Figure C and D should be changed to Table.

Figure 5A, B

Since the X scale is not same between A and B, it is difficult to compare the result. Authors should adjust X scale of 5B.

Author Response

Dear reviewer,

We are very appreciated for the critical and detailed comments,

here are our responses:

Point 1: Abstract, L68: Please define the meaning of POD.

Response 1: We have defined the meaning of POD in abstract.

Point 2: Figure 3B, Figure 4, L209, L442-445:

Figure 3B suggest that relative BcXyl activity is highest in pH 5.0, and low in pH7.0.

However, in anthocyanin degradation activity assay in Figure 4, assay was performed at pH7.0. Why did authors perform the assay in this condition? In addition, since “Spontaneous” anthocyanin degradation is high in this assay, it is recommended to consider assay condition.

Response 2: It is true that the optimum pH we detected for BcXyl on pNPGa was pH 5.0. Actually, we started to measure the activity of BcXyl on Brunflesia anthocyanins at pH 5.0, but we could not get the activity. So, we detected the activity at a series of pH 3.0, 4.0, 5.0, 6.0, and 7.0. We could only get the activity at pH 7.0. It is known that the impact of acid condition (pH) on anthocyanin is significant. The color, stability and molecular structure of anthocyanins are dependent on pH.  At low pHs, the molecules exist as flavylium cation. At pH 5.0, they exist as pseudobases, while at pH 7.0 as chalcones. We believe the molecular structure may influence the binding of the substrate for BxXyl. At pH 7.0, anthocyanins are not stable as they are at low pHs. Accordingly, non-enzymatic degradation or “spontaneous” degradation may occur. We knew that this might interfere the activity assay. And we did see the interference in the activity assay by HPLC (Figure 4) and by GC/MS (Figure 5). This is the reason we set up denatured enzyme control in parallel to rule out the interference. Under the assay condition and subtraction of the data of the control, we detected the pigment degradation or release of sugar, indicating the activity of BcXyl on the anthocyanins.

Point 3: Figure 3: Figure C and D should be changed to Table.

Response 3: We agree with the reviewer and we already changed Figure 3C and D to Tables.

Point 4: Figure 5A, B: Since the X scale is not same between A and B, it is difficult to compare the result. Authors should adjust X scale of 5B.

Response 4: We agree with the reviewer and we already adjusted the scale.

Else, we also submit our responses via a word document, please see the attachment file.

Best regards,

Xuequn Pang and Zhaoqi Zhang

Round  2

Reviewer 1 Report

Thank you for revising the manuscript. 

I am sorry, but I still have the same question than before concerning Figure 2B: 

The fractions collected as indicated in (A) was further separated by CM‐Sepharose column 

chromatography and fractions with high activity while low protein content (Fraction No. 26‐28) were collected for purity analysis.

The authors have added some material and methods information (not necessary) but I still do not understand why the authors explain that the protein content is LOW. In the figure it looks like the absorption at 280 nm increases with the activity which means that the protein content is HIGH compared to the other fractions. Did the authors analyze the fractions and observe other compounds absorbing at 280 nm explaining the increase or is there a mistake in the figure or in the text?

Author Response

Dear reviewer,

Thanks again for the critical comments. Here are our response:

So sorry for the careless mistake in the text, we forgot to fixed this in the legend, now we checked and fixed both the result part and the legend in Figure 2. 

Best regards,

Xuequn Pang

Reviewer 2 Report

Authors revised and improved manuscript and responded clearly for question. 

Since response 2 shown as below is very informative for readers, please consider adding discussion of impact of acid condition (pH) on molecular structure of anthocyanins and BcXyl activity.

Response 2

It is true that the optimum pH we detected for BcXyl on pNPGa was pH 5.0. Actually, we started to measure the activity of BcXyl on Brunflesia anthocyanins at pH 5.0, but we could not get the activity. So, we detected the activity at a series of pH 3.0, 4.0, 5.0, 6.0, and 7.0. We could only get the activity at pH 7.0. It is known that the impact of acid condition (pH) on anthocyanin is significant. The color, stability and molecular structure of anthocyanins are dependent on pH. At low pHs, the molecules exist as flavylium cation. At pH 5.0, they exist as pseudobases, while at pH 7.0 as chalcones. We believe the molecular structure may influence the binding of the substrate for BxXyl. At pH 7.0, anthocyanins are not stable as they are at low pHs. Accordingly, non-enzymatic degradation or “spontaneous” degradation may occur. We knew that this might interfere the activity assay. And we did see the interference in the activity assay by HPLC (Figure 4) and by GC/MS (Figure 5). This is the reason we set up denatured enzyme control in parallel to rule out the interference. Under the assay condition and subtraction of the data of the control, we detected the pigment degradation or release of sugar, indicating the activity of BcXyl on the anthocyanins.

Author Response

Dear reviewer, 

Thanks again for your critical comments. Here is our response:

Point 1:

Since response 2 shown as below is very informative for readers, please consider adding discussion of impact of acid condition (pH) on molecular structure of anthocyanins and BcXyl activity.

Response 1:

We agree with the reviewer and already added the proper discussion in “3.2”.

Best regards,

Xuequn Pang